# An explainable multi-objective hybrid machine learning model for reducing heart failure mortality



F M Javed Mehedi Shamrat[1], Majdi Khalid[2], Thamir M. Qadah[3], Majed Farrash[2] and Hanan Alshanbari[2]

[1] Department of Computer System and Technology, Universiti Malaya, Kuala Lumpur, Malaysia
[2] Department of Computer Science and Artificial Intelligence, Umm Al-Qura University, Makkah, Saudi Arabia
[3] Department of Computer and Network Engineering, Collge of Computing, Umm Al-Qura University, Makkah, Saudi Arabia

## ABSTRACT

As the world grapples with pandemics and increasing stress levels among individuals, heart failure (HF) has emerged as a prominent cause of mortality on a global scale. The most effective approach to improving the chances of individuals' survival is to diagnose this condition at an early stage. Researchers widely utilize supervised feature selection techniques alongside conventional standalone machine learning (ML) algorithms to achieve the goal. However, these approaches may not consistently demonstrate robust performance when applied to data that they have not encountered before, and struggle to discern intricate patterns within the data. Hence, we present a Multi-objective Stacked Enable Hybrid Model (MO-SEHM), that aims to find out the best feature subsets out of numerous different sets, considering multiple objectives. The Stacked Enable Hybrid Model (SEHM) plays the role of classifier and integrates with a multi-objective feature selection method, the Non-dominated Sorting Genetic Algorithm II (NSGA-II). We employed an HF dataset from the Faisalabad Institute of Cardiology (FIOC) and evaluated six ML models, including SEHM with and without NSGA-II for experimental purposes. The Pareto front (PF) demonstrates that our introduced MO-SEHM surpasses the other models, obtaining 94.87% accuracy with the nine relevant features. Finally, we have applied Local Interpretable Model-agnostic Explanations (LIME) with MO-SEHM to explain the reasons for individual outcomes, which makes our model transparent to the patients and stakeholders.

## INTRODUCTION

Heart failure (HF) is a medical illness marked by the heart's incapacity to effectively fulfill the body's metabolic requirements (*Sutradhar et al., 2023b*). Even with significant progress in medical technology, HF remains prevalent and is a leading cause of death worldwide (*World Health Organization, 2023*). Excessive alcohol consumption, sedentary lifestyle, smoking, chest radiation, obesity, infectious agents, influenza, hypertension, and

Corresponding author
F M Javed Mehedi Shamrat,
javedmehedicom@gmail.com

dyslipidemia are common contributors to HF (*McDonagh et al., 2021*). Additionally, age, family medical history, gender, and other non-lifestyle factors also have the potential to play a role. Compared to men and younger people, women and older people are more vulnerable (*Peters et al., 2021*; *Tromp et al., 2021*). It is estimated that the prevalence of HF is expected to rise by approximately 46% between 2012 and 2030, accompanied by the death rates (*Shahim et al., 2023*).

Promptly assessing mortality indicators and initiating treatment with counseling and medications is vital for reducing the death rate. To determine the risk of HF mortality, conventional measures like renal function (a sign of poor kidney function), B-type natriuretic peptide (a hormone released by the heart in response to HF), ejection fraction (a measure of heart pumping efficiency), and various clinical parameters are used (*Sutradhar et al., 2023c*). However, the conventional method can be quite complex, time-consuming, and expensive, and it might not always produce the desired results. Consequently, researchers have focused on leveraging machine learning (ML) techniques to explore indicators of HF mortality. ML plays a significant role in hybrid models by handling complex patterns, adapting over time, and improving decision-making abilities (*Khalid et al., 2023*). This synergy makes hybrid models particularly useful for solving complex, real-world problems (*Talukder et al., 2023*).

Additionally, multi-objective optimization (*Kalita et al., 2023*) and explainable artificial intelligence (AI) (*Khan et al., 2023*) can be utilized to produce precise, comprehensible, and therapeutically useful models for forecasting HF mortality. In order to ensure strong models that capture a variety of parameters while staying manageable for clinical usage, multi-objective optimization strikes a compromise between accuracy, simplicity, and feature selection (*Sutradhar, Bidgoli & Rahnamayan, 2024*). Clinicians can better grasp feature significance and causal patterns with the help of explainable AI, which offers transparency into the model's decision-making process (*Yilmaz et al., 2024*). These strategies promote trust, and make data-driven, morally good decisions possible, which eventually enhances patient outcomes and encourages clinical adoption.

As ML has evolved, the scope of features utilized has expanded to encompass thousands or even millions of variables, and the significance of feature selection techniques has become increasingly apparent. Eliminating irrelevant or redundant features not only decreases computational expenses, but also enhances the classifier's effectiveness, while also rendering the model more interpretable (*Sutradhar et al., 2023a*). Consequently, conventional or supervised feature selection techniques are broadly used in ML-based HF diagnosis before entering the predictive task. For example, *Newaz, Ahmed & Haq (2021)* used two well-known feature selection techniques named Chi-square and Recursive Feature Elimination to predict the mortality of HF. In parallel, *Ishaq et al. (2021)* employed the feature importance-based feature selection method in the Faisalabad Institute of Cardiology (FIOC) dataset. Additionally, the wrapper-based feature selection method was utilized in the study in *Le et al. (2021)*. Afterward, *Hussain et al. (2021)* applied both wrapper and filter-based feature selection methods for their study. However, the broad dependence on supervised feature selection methods has several drawbacks, including

limited consideration of feature interactions, inadequate handling of feature redundancy, and lack of optimization (*Rabash et al., 2023*; *Nguyen, Xue & Zhang, 2020*).

As a result, instead of using these conventional methods, we have turned our attention to applying multi-objective optimization, named Non-dominated Sorting Genetic Algorithm II (NSGA-II), aiming to achieve the highest accuracy considering all of the different features set at the same time. It generates a set of solutions, where each solution represents a trade-off between different objectives and allows us to explore the trade-offs between different objectives and select the most suitable solution based on our requirements (*Verma, Pant & Snasel, 2021*). NSGA-II addresses the limitations of conventional selection methods by considering multiple conflicting objectives, and efficiently searching for diverse and high-quality solutions in complex feature spaces (*Jiao et al., 2023*). Nevertheless, in the context of HF mortality, no existing studies are focused on it.

Some studies have applied it in the realm of other areas. For instance, *Soui et al. (2021a)* applied NSGA-II as a feature selection and used the AdaBoost (AB) as a classifier. Additionally, *Gupta et al. (2022)* utilized Extreme Gradient Boosting, known as XGB as a performance indicator. Another traditional classifier K Nearest Neighbors or KNN is integrated with NSGA-II in the study of *Zhu & Neri (2023)*. However, these traditional classifiers may perform well when trained and tested on the same set of features that were selected by NSGA-II. This can lead to biased performance estimates, as the classifier may overfit the selected features and perform poorly on unseen data. Therefore, we initially trained five different conventional ML classifiers, and then based on their performance, we chose the top-performing classifier and proposed a Stacked Enable Hybrid Model (SEHM). By aggregating the prediction of a top-performing ML classifier, an ensemble method can better capture complex patterns and reduce overfitting (*Sutradhar et al., 2023b*). After that, we used SEHM as a performance indicator of NSGA-II, providing more reliable and informative evaluations of the feature selection process. Finally, we performed local interpretable model-agnostic explanations (LIME) to interpret our proposed model by shedding light on how it reaches its outcomes. Our primary contributions are as follows:

- Address data imbalance before implementing the ML model using Synthetic Minority Over-sampling Technique (SMOTE) and TomekLinks. These techniques produce synthetic instances to address class imbalance and eliminate noisy or irrelevant instances if detected.

- Presented a Multi-objective Stacked Enable Hybrid Model (MO-SEHM) to mitigate the drawbacks of supervised feature selection techniques. The MO-SEHM combines a multi-objective feature selection method NSGA-II and SEHM as a classification method.

- Pareto front (PF) and other performance indicators demonstrate that our proposed MO-SEHM achieves the highest result 94.87% accuracy with nine relevant features out of 12.

- Finally, to make our proposed model transparent and enhance the trust ability issues to patients and stakeholders, we consider applying an explainable AI method named LIME to the MO-SEHM. This provides the reasons why our model reaches the predictive decision and gives confidence in the diagnosis application.

The structure of the article unfolds as follows: "Related Work" delves into the existing literature, while "Research Methodology" elucidates the systematic research methodology adopted in this study. Subsequently, "Experimental Analysis and Discussion" offers insights into the experimental outcomes, analysis, and discussions. The article culminates with conclusions drawn in "Discussion".

## RELATED WORK

Recent studies have extensively explored the use of machine learning techniques to effectively predict heart failure mortality. For instance, *Newaz, Ahmed & Haq (2021)* employed various machine learning classifiers such as Random Forest (RF), KNN, AB, and Support Vector Machine, known as SVM to assess the risk of HF mortality. Their results indicated that RF outperformed other classifiers with an accuracy of 76.25%, particularly when utilizing Chi-square-based feature selection. Similarly, *Li et al. (2022)* aimed to construct a predictive model using machine learning, with the XGB classifier achieving the highest performance, yielding an area under the curve (AUC) of 82.4%. *Ishaq et al. (2021)* determined the relevant features of HF mortality by applying the feature importance-based feature selection techniques and addressed imbalance issues using SMOTE. Among the classifiers tested, Extra Trees (ET) achieved the highest accuracy of 92.62%. *Mishra (2022)* and *Plati et al. (2021)* also addressed imbalance issues through SMOTE and found that Rotation Forest Tree (ROT) and SVM classifiers achieved the highest accuracies of 91.3% and 83.33%, respectively. *Nishat et al. (2022)* examined the UCI HF dataset using six conventional ML classifiers. The RF classifier performed better than the others, with 90% accuracy when Synthetic Minority Over-sampling Technique with Edited Nearest Neighbors (SMOTE-ENN) was used to address the imbalance problem. However, these aforementioned studies rely on a single model that limits accuracy, robustness, and adaptability, making hybrid models a more effective solution for complex tasks (*Sutradhar et al., 2024*). The proposed SEHM, a hybrid ML model offers improved accuracy by combining the strengths of multiple algorithms, while reducing individual weaknesses. They enhance robustness, ensuring reliable performance even if one component underperforms. Additionally, hybrid models can improve scalability and adaptability, making them effective across diverse datasets and changing environments.

*Le et al. (2021)* later applied the Grey Wolf Optimization feature selection method and compared the outcomes from seven ML classifiers. Their findings revealed that RF produced an optimal accuracy of 85%. *Hussain et al. (2021)* utilized a range of ML classifiers, with SVM demonstrating superior overall performance with an accuracy of 88.79% when using all multimodal features. *Sabahi, Vali & Shafie (2023)* and *Luo et al. (2022)* used the XGB classifier to obtain 76.4% accuracy and 83.1% AUC, respectively, by applying feature importance-based selected features. Serum creatinine and ejection fraction were the only two patient parameters (*Chicco & Jurman, 2020*) used to predict the survival of HF patients. Their predictive model with the RF classifier yielded a 74% accuracy rate. In a separate study (*Mpanya et al., 2023*), six supervised machine learning classifiers were used to create a model for predicting hospital mortality in HF patients. The authors claim that RF had the highest accuracy of 88% during the testing phase. These

research, however, used conventional feature selection methods, which concentrate on a particular goal (such as increasing accuracy or lowering feature count). This might result in unbalanced solutions, such as high accuracy with poor interpretability or excessive complexity (*Sutradhar, Bidgoli & Rahnamayan, 2024*). A multi-objective feature selection, such as NSGA-II, solves this by simultaneously optimizing multiple conflicting objectives, ensuring a better trade-off between accuracy, interpretability, and efficiency. This helps in building more robust, generalizable, and practical models.

As consequently, some research concentrated on multi-objective optimization, likewise, *Muthulakshmi, Kavitha & Aishwarya (2022)* classified HF rankings using a nature-inspired butterfly optimization algorithm (BOA). They showed a better result using SVM with 93.1% accuracy and suggested utilizing a multi-objective approach prior to sending data in the classification phase. *Gupta et al. (2022)* classified people with and without diabetes using NSGA-II and XGB. The sensitivity of their suggested model, which used a hybridized dataset, was 96.36%. Another research (*Gupta & Singh, 2023*) uses a number of stand-alone machine learning models, including a KNN, SVM, Bayesian Belief Networks (BBN), RF, and Naive Bayes (NB), to predict heart disease. The accuracy, sensitivity, specificity, precision, and F-measure of the model that was provided were 97.32%, 92.84%, 92.60%, 91.25%, and 92.17%, respectively. Further, *Sutradhar, Bidgoli & Rahnamayan (2024)* used a variety of datasets in the context of electroencephalography (EEG) and showed that combining RF with NSGA-II produced the best results.

Several researchers have introduced hybrid ensemble models in their studies. For instance, *Mohan, Thirumalai & Srivastava (2019)* combined the RF classifier with a linear model to create a hybrid model named Hybrid Random Forest with a Linear Model (HRFLM), which achieved 88.7% accuracy. *Rahman et al. (2023)* proposed a hybrid model using Stacking (ST) by combining three baseline classifiers. Their proposed model surpassed single traditional ML classifiers and achieved an accuracy of 89.41%. Another hybrid model, proposed by *Sutradhar et al. (2023c)*, is named Combining the Best Classifier with an Ensemble Classifier (CBCEC) and is developed by combining multiple ensemble methods. This proposed hybrid model achieved 93.67% accuracy with the FIOC dataset. Additionally, *Ghosh et al. (2021)* introduced some hybrid ML models by integrating single traditional and ensemble classifiers. They individually used baseline classifiers such as Decision Tree (DT), Gradient Boost (GB), RF, KNN, and AB as base estimators of the ensemble method. However, considering the superiority of multi-objective feature selection and hybrid ML classifier, we aim to integrate NSGA-II and SEHM to ensure balanced feature selection, optimizing trade-offs between accuracy, interpretability, and efficiency. This improves generalization, robustness, and scalability by selecting relevant features, reducing overfitting, and minimizing computational cost (*Rashid et al., 2024*). The combination offers a flexible and adaptive solution for complex real-world tasks.

Furthermore, *Moreno-Sánchez (2023)* uses SHapley Additive exPlanations (SHAP), an explainable AI, to explain the results given RF with 74% accuracy in order to make his research visible to patients and stakeholders. Serum creatinine (Ser-C), ejection fraction (Ej-Fr), and sex were identified as the most significant characteristics for predicting heart

failure mortality in their suggested model. Subsequently, *Wrazen et al. (2023)* introduced an explainable model called DeepSHAP, in which they combine DT as a classifier with SHAP to explain DT results. Their suggested approach produced exceptional area under the receiver operating characteristic (AUC-ROC) values from various feature sets, about 83%. Finally, *El-Sofany, Bouallegue & El-Latif (2024)* used five machine learning methods to predict cardiac disease: SVM, XGB, Bagging (BG), DT, and RF. They combine SHAP with the top-performing classifier, XGB, to learn more about how the system makes its final predictions. Their persistent emphasis on SHAP, however, may limit the clinical usefulness of their suggested models due to its complexity, particularly when combined with a hybrid model (*Wang et al., 2024*). LIME, another explainable AI, provides faster, locally focused, and model-agnostic explanations that successfully emphasize patient-specific insights, making it better in scenarios where speed and interpretability are critical (*Shtayat et al., 2023*). Its feature restriction features and ease of deployment make it beneficial in clinical situations. These studies are explicitly compared in Table 1. The symbol (✗) signifies that this research is not regarded as a particular methodology or approach.

## RESEARCH METHODOLOGY

In this section, we comprehensively discuss the methodologies and procedures employed in the research. The operational methodology is divided into five main components, which include data collection, data preprocessing, model deployment, objective optimization, and interpretable model. Figure 1 illustrates an overview of the overall working process.

### Data collection and preprocessing

This research utilized the HF clinical records dataset from the FIOC, which is now publicly accessible in *Kaggle (2023)*. During the April to December 2015 follow-up period, the dataset comprised 299 patients with heart problems, including 194 men and 105 women. Their ages ranged from 40 to 95 years, and all 299 patients had left ventricular systolic dysfunction and prior heart failures, categorizing them under stages III or IV of the New York Heart Association (NYHA) classification for heart failure. The average follow-up duration was 130 days, with a minimum of 4 days and a maximum of 285 days. Table 2 represents the features, range of value with measurements, and their corresponding explanations of the employed dataset. Some features exhibit binary characteristics, such as diabetes, anemia, smoking, sex, high blood pressure (HBP), and death-event. The remaining features encompass a mixture of integer and floating-point characteristics. For classification purposes, the death-event feature was chosen as the target variable, indicating whether the patient survived or died (where one represents deceased and 0 represents survived) before the end of the follow-up period, with 203 reported deceased cases and 96 survivors, illustrating an imbalance sign.

In our experimentation, dealing with imbalanced datasets has emerged as a significant challenge. This may cause the model to be biased in favor of the majority class, which would be bad for the minority class and lead to inaccurate evaluation measures (*Thabtah et al., 2020*). Thereby, academics are concerned about this problem and want to solve it

**Table 1  An explicit comparison between recent ML-based studies on heart failure mortality.**

| Author and year | Data-balancing technique | Feature optimization | Performed model | Explainable AI |
|---|---|---|---|---|
| *Newaz, Ahmed & Haq (2021)* | SMOTE | Single-objective | RF | (✗) |
| *Ishaq et al. (2021)* | SMOTE | Single-objective | ET | (✗) |
| *Le et al. (2021)* | (✗) | Single-objective | RF | (✗) |
| *Hussain et al. (2021)* | (✗) | Single-objective | SVM | (✗) |
| *Sutradhar et al. (2023b)* | SMOTE-ENN | Single-objective | IBS | (✗) |
| *Li et al. (2022)* | (✗) | Single-objective | XGB | SHAP |
| *Mishra (2022)* | SMOTE | Single-objective | SVM | (✗) |
| *Plati et al. (2021)* | SMOTE | Single-objective | ROT | (✗) |
| *Nishat et al. (2022)* | SMOTE-ENN | Single-objective | RF | (✗) |
| *Sabahi, Vali & Shafie (2023)* | SMOTE | Single-objective | XGB | (✗) |
| *Luo et al. (2022)* | (✗) | Single-objective | XGB | (✗) |
| *Chicco & Jurman (2020)* | (✗) | Single-objective | RF | (✗) |
| *Mpanya et al. (2023)* | (✗) | Single-objective | RF | (✗) |
| *Mohan, Thirumalai & Srivastava (2019)* | (✗) | Single-objective | HRFLM | (✗) |
| *Rahman et al. (2023)* | (✗) | Single-objective | Stacking | (✗) |
| *Sutradhar et al. (2023c)* | Consisting of Boosting, SMOTE, and Tomek links (BOO-ST) | Single-objective | CBCEC | (✗) |
| *Muthulakshmi, Kavitha & Aishwarya (2022)* | (✗) | Multi-objective | SVM | (✗) |
| *Moreno-Sánchez (2023)* | (✗) | Single-objective | RF | SHAP |
| *Wrazen et al. (2023)* | (✗) | Single-objective | DT | SHAP |
| **Our study** | **SMOTE and Tomeklink** | **Multi-objective** | **SEHM** | **LIME** |

before model training. SMOTE was selected to address the class imbalance by generating synthetic instances for the minority class, as it is widely favored by researchers in medical datasets (*Newaz, Ahmed & Haq, 2021*; *Ishaq et al., 2021*; *Mishra, 2022*; *Plati et al., 2021*). Let the feature matrix be $X$ and the target vector is $Y$, where $X$ contains $m$ instances and $n$ features, and $Y$ contains the corresponding level of each instance. SMOTE works by selecting a minority class instance $x_m$ and its k-nearest neighbors ($k > 0$). A new synthetic instance $x_n$ is created by interpolating between $x_m$ and one of its nearest neighbors, and this procedure is repeated until the desired balance between classes is achieved. However, to make the dataset balance by generating synthetic instances, this procedure frequently introduces irrelevant and noisy data (*Cheng et al., 2019*). Hence, we applied Tomeklink in the generated synthetic instances to identify and remove instances that are difficult to classify or noisy. Let $x_i$ and $x_j$ be two instances belonging to different classes, and $d(x_i, x_j)$ be the distance between them. If there exists no other instance $x_k$ such that $d(x_i, x_k) < d(x_i, x_j)$ or $d(x_j, x_k) < d(x_i, x_j)$, then $x_i$ and $x_j$ form a Tomeklink. Removing Tomeklinks helps improve the separation between classes by eliminating borderline or noisy instances.

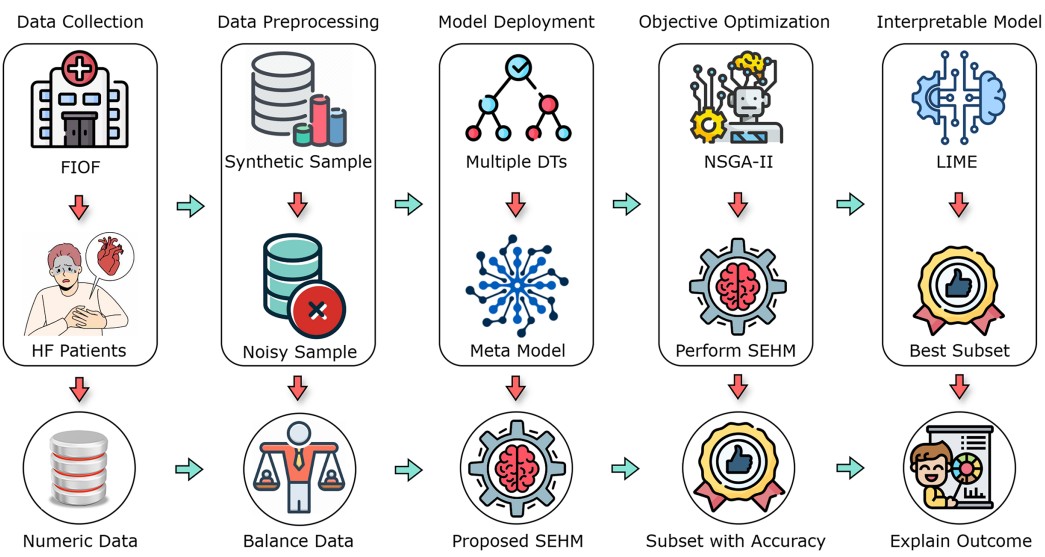

Data Collection   Data Preprocessing   Model Deployment   Objective Optimization   Interpretable Model

FIOF → HF Patients → Numeric Data   |   Synthetic Sample → Noisy Sample → Balance Data   |   Multiple DTs → Meta Model → Proposed SEHM   |   NSGA-II → Perform SEHM → Subset with Accuracy   |   LIME → Best Subset → Explain Outcome

**Figure 1  A flow diagram of our proposed study with different primary components.**

**Table 2  Explanations of the employed HF dataset.**

| Variables | Value range | Measurement | Explanation |
| --- | --- | --- | --- |
| Age | 40–95 | Year | Patient age in years |
| Anaemia | 0 (no), 1 (yes) | Boolean | Decrease of hemoglobin or blood cells |
| High blood pressure (HBP) | 0 (no), 1 (yes) | Boolean | Whether the patient has blood pressure |
| Creatinine phosphokinase (Cr-Ph) | 23–7,861 | Mgc/L | Level of the CPK enzyme in the blood |
| Diabetes | 0 (no), 1 (yes) | Boolean | Whether the patient has diabetes Boolean |
| Ejection fraction (Ej-Fr) | 14–80 | Percentage | Blood leaving percentage |
| Sex | 0 (woman), 1 (man) | Binary | Whether the patient is man or woman |
| Platelets | 25.01–850.00 | Kilo platelets/mL | Platelets in the patient blood |
| Serum creatinine (Ser-C) | 0.50–9.40 | mg/dL | Level of creatinine in the patient blood |
| Serum sodium (Ser-S) | 114–148 | mg/dL | Level of sodium in the patient blood |
| Smoking | 0 (no), 1 (yes) | Boolean | Whether the patients smoke or not |
| Time | 4–285 | Days | Patients follow-up period |
| Death-event | 0 (survived), 1 (dead) | Boolean | Whether the patient died in the follow-up period |

Afterward, the balanced dataset was split by using five-fold stratified cross-validation (SCV). This strategy can improve the model's generalizability and accuracy by assuring that every fold has a representative sample for testing and training over an equal number of classes. In this way, each fold serves as the test set once during the five iterations of the models, while the other four folds are used for training. By averaging the model's performance across several folds, we can obtain a more accurate and trustworthy evaluation of its performance on unknown data (*Sutradhar et al., 2023a*).

**Traditional ML classifiers**

Once data was prepared, we applied five widely recognized traditional ML classifiers, including Decision Trees (DT), Support Vector Machines (SVM), Extra-Tree (ET), K-Nearest Neighbors (KNN), and Random Forest (RF). A concise overview of each classifier is provided in the following sections.

### Decision Tree (DT)

DT employs an algorithmic approach to construct multiple DTs by determining the optimal criteria. This process known as splitting, involves selecting the best feature to build the tree from the root node upwards. The identification of the best feature is based on calculations of entropy ($E$) and information gain ($G$). The formulas for $E$ and $G$ are provided in Eqs. (1) and (2), where $X$ represents the attributes, $Y$ denotes the class level, and ($P+$) and ($P-$) indicate positive and negative samples, respectively (*Kotsiantis, 2013*).

$$E(D) = -(P+)log_2(P+) - (P-)log_2(P+), \tag{1}$$
$$G(X) = E(X) - E(X, Y). \tag{2}$$

### Support Vector Machine (SVM)

SVM is a powerful supervised learning method that operates by determining the optimal decision boundary or hyperplane (*Ding et al., 2021*). The functional representation of SVM is presented in Eq. (3), where $X$ denotes the input, $W$ represents the weight, $B$ indicates the bias, $T$ signifies the transpose operation, and $SIGN()$ is a function that yields either $1+$ or $1-$ based on the input data type.

$$SVM(X) = SIGN\{W^{(T)}X + B\}. \tag{3}$$

### Extra Tree (ET)

The ET is a form of ensemble learning that enhances model accuracy by generating multiple DTs and combining their predictions. ET randomly selects split points, creating numerous DTs, each with random split points for every feature. The mathematical formulation is illustrated in Eq. (4), where $E(Y)$ represents the predicted outcome, $n$ is the total number of DTs, and $w_i$ and $h_i$ denote the weight and predicted output of the $i^{th}$ tree, respectively, for the input $X$.

$$E(Y) = \sum_{i=0}^{n} \{w_i h_i(X)\}. \tag{4}$$

### K Nearest Neighbors (KNN)

KNN is designed to determine the optimal class for test data by evaluating the distance between the test data and training points. The fundamental operation of KNN involves calculating the Euclidean distance in Eq. (5) between each pair of raw training data and the test data. In this equation, $(X_1, X_2)$ and $(Y_1, Y_2)$ represent the coordinates of the first and second points, respectively (*Muhammad et al., 2023*).

$$Euclidean = \sqrt{(X_2 - X_1) + (Y_2 - Y_1)}. \tag{5}$$

### Random Forest (RF)

The RF is an ensemble of DTs, each tree within the ensemble is created from a sample drawn from the training set with replacement. The features are represented as $X = \{x_1, ..., x_n\}$ with corresponding class level $Y = \{y_1, ..., y_n\}$, where $n$ denotes the number of samples. The index $l$ ranges from a minimum value of 1 to an upper limit of $L$ and predictions are made by averaging predictions for $x^p$ provided by each tree for $x$, as illustrated Eq. (6) (*Ghosh et al., 2021*).

$$RF = \frac{1}{L}\sum_{l=1}^{L} l(x^p). \tag{6}$$

## Multi-objective Stacked Enable Hybrid Model (MO-SEHM)

In this section, we present our proposed MO-SEHM, designed to address issues associated with conventional feature selection techniques and standalone traditional ML classifiers. As previously mentioned, we introduce a Stacked Ensemble Hybrid Model (SEHM) to mitigate the drawbacks of standalone ML classifiers, including bias and overfitting, and to enhance overall performance. Additionally, prior studies have predominantly relied on supervised feature selection techniques, where a feature set is randomly selected and fitted into a predictive model. However, there is a high likelihood that other subsets may prove more promising and effective, as these were not thoroughly evaluated. Hence, we advocate for the application of a multi-objective feature selection method, NSGA-II, which simultaneously considers multiple objectives and evaluates each potential feature subset using a performance metric. Finally, we integrated SEHM as a performance indicator of our proposed method, working steps are outlined in Algorithm 1.

### Initial stage: stacked enable hybrid model

In recent times, researchers have increasingly focused on crafting hybrid classifiers to address the limitations inherent in single classifiers. Unlike a single traditional classifier, which might struggle with certain data types and have limited capacity to encompass various aspects of the dataset, hybrid classifiers leverage the combined capabilities of multiple classifiers. Each classifier contributes unique perspectives to the analysis, resulting in a final hybrid classifier that delivers more resilient and comprehensive results. However, since the hybrid model combines the outputs of multiple classifiers, including one with poor performance may negatively impact the final predictions. This could lead to inaccurate or less reliable results compared to using only higher-performing classifiers in the hybrid model. Hence, we have made a comparative evaluation between the employed traditional classifiers and chosen the top-performing model (TPF) based on their outcomes. Equation (7) illustrates the procedures, where $X_{test}$ indicates the testing data and $Max_{acc}$ refers to considering the maximum accuracy from these classifiers.

---

**Algorithm 1  MO-SEHM.**

1:  **Inputs:** Dataset $D = \sum_{i=1}^{N} (X_i, Y_i)$.

2:  **Outputs:** Optimal solution for HF mortality.

3:  *Initial-Phase:* Develop the SEHM.

4:  $DT_m \leftarrow$ Generate the number decision tree

5:  **for** $i \leftarrow 1; i \leq m;$ i ++ **do**

6:      $B_s \leftarrow$ bootstrap samples from $D_{train}$

7:      $DT_i \leftarrow$ built decision trees

8:      $BP_i \leftarrow$ producing the base predictions

9:  **end for**

10:  **for** $j \leftarrow 1; j \leq f;$ k++ **do**

11:      $MF_j \leftarrow$ generating the meta-features using $BP$

12:      $SEHM \leftarrow$ evaluate $MF$ using $LG$

13:  **end for**

14:  *Final-Phase:* Apply NSGA-II and integrate SEHM.

15:  Initialize parameters: population $(P)$, generations $(G)$,…,

16:  $L \leftarrow$ length of input features

17:  **for** $l \leftarrow 1; l \leq L;$ l++ **do**

18:      **for** $g \leftarrow 1; g \leq G;$ g++ **do**

19:          **for** $p \leftarrow 1; p \leq P;$ p++ **do**

20:              $Fitness_p \leftarrow SEHM\left(X_p^{(SEF)}\right)$

21:          **end for**

22:          $NextGEN \leftarrow NSGAII(P, Fitness)$

23:      **end for**

24:  **end for**

25: $PF \leftarrow IdentifyPF(NextGEN, Fitness)$

26: *Optimal Solution* $\leftarrow$ *Choose Optimal Solutation(PF)*

---

$$TPF = Max_{acc}\{DT(X_{test}), \ldots, RF(X_{test})\}. \tag{7}$$

Based on the experiments, we concluded that RF classifiers outperformed others in terms of predictive results. Hence, as per the working procedure of RF, we randomly select $m$ features from the dataset and create bootstrap samples $B_s$. Recursively splitting the dataset by $B_s$, build the $m$ number of decision trees $DT_m$, then train each $DT_m$ using the training data $(D_{train})$ and get the base prediction. Subsequently, to strengthen our approach and improve the final outcome, we generate the $(j = 1, 2, \ldots, f)$ number of meta-features (MF) using the first-level base predictions. Finally, a meta-model Logistic Regression (LG) is employed to evaluate the $f^{th}$ meta-features, expressed in Eq. (8). This two-level stacked

ensemble method can help to reduce the impact of noisy or incorrect predictions and improve the generalization of the predicted outcome for our model.

$$SEHM = \sum_{j=1}^{f} \{LG(MF)\}. \tag{8}$$

### Final stage: multi-objective feature selection

Multi-objective optimization encompasses the optimization of two or more conflicting objectives concurrently, posing a challenge in selecting an appropriate metric for evaluating candidate solutions. Algorithms tailored for such problems typically navigate a trade-off decision-making process, yielding a set of solutions instead of a single outcome. A multi-objective problem can be formulated as described in Eq. (9) and (10).

$$F(X) = MIN/MAX\{f_1(X), f_2(X), ..., f_M(X)\}. \tag{9}$$

Subject to,

$$\begin{aligned} g_j(X) &\geq 0, & j &= 1, 2, ..., J, \\ h_k(X) &= 0, & k &= 1, 2, ..., K, \\ x_i^{(L)} \leq x_i &\leq x_i^{(U)}, & i &= 1, 2, ..., N. \end{aligned} \tag{10}$$

Here $f_m$ is the $m^{th}$ objective of $F(X)$, where, $m = 1, 2, ..., M$; $g_j(X)$ is $j^{th}$ inequality constraint; $h_K$ is $k^{th}$ equality constraint; $X = (x_1, x_2, ..., x_N)$ is a $N$-dimensional vector; finally, $x_i^{(L)}$ and $x_i^{(U)}$ are the lower and upper bounds on $i^{th}$ variable. Additionally, dominance plays a pivotal role in multi-objective optimization by comparing and ranking candidate solutions, facilitating the identification of non-dominated solutions (NDSS). The solutions forming the Pareto front, unmatched by any other solution, are termed NDSS. Solution $X^{(1)}$ dominates solution $X^{(2)}$ if $X^{(1)}$ is at least as good as $X^{(2)}$ across all objectives, and $X^{(1)}$ is strictly superior to $X^{(2)}$ in at least one objective. These conditions are mathematically expressed as follows,

$$\begin{aligned} f_j\{X^{(1)}\} \not\vartriangleright f_j\{X^{(2)}\}, & \quad \forall_j = 1, 2, \cdots, M; \wedge, \\ f_{j^+}\{X^{(1)}\} \vartriangleleft f_{j^+}\{X^{(2)}\}, & \quad \exists \forall_{j^+} \in \{1, 2, \cdots, M\}. \end{aligned} \tag{11}$$

Here, we utilize the popular multi-objective optimization method, NSGA-II on the set of 12 input features, specifying the population accordingly and computing objective values for each individual. The PF is found using the Non-Dominated Sorting (NDS) algorithm once objective values for the beginning population have been calculated. Based on how well each solution performs, its fitness is evaluated using the SEHM. The next steps entail using the best candidate solutions, which makes dominant candidates superfluous and removes them from the population. Within the PF solutions, new solutions are created iteratively, and their related objective values are determined. After comparing these recently created solutions to their corresponding parents, any that are determined not to be dominated by their parents are added to a temporary list. The usefulness of each feature is

methodically examined in this sequential examination. Eventually, those on the temporary list are integrated into the general population.

The modified population is subjected to the NDS algorithm once again to determine the PF; any solutions that remain are eliminated. It is crucial to confirm the population size at this point. The crowding distance is used to choose the *P* best solutions for the PF, which eliminates extra people from the population, if the number of members in the PF surpasses a predetermined threshold (such as the length of *P*). Ultimately, we identify the best course of action by extracting the PF based on the fitness of the existing population.

# EXPERIMENTAL ANALYSIS AND DISCUSSION

This section presents a thorough analysis of the experimental outcomes derived from our proposed methodology. A precise set of parameters is used to train each classifier in order to regulate the learning process. We employed GridSearchCV in this context, which finds the best model configuration by methodically examining the hyperparameter space and automating the hyperparameter tweaking process (*Sutradhar et al., 2024*). The used optimal parameters for various feature sets are shown in Table 3. We have structured our experimentation into two segments: initially assessing performance outcomes without the integration of NSGA-II, followed by the evaluation incorporating NSGA-II. To ensure a comprehensive evaluation, we have examined various classification metrics, including accuracy, precision, recall, F1-score, ROC curve, Jaccard score, Cohen's Kappa score, and Hamming loss. Furthermore, for the NSGA-II integrated section, we consider the Pareto front. Lastly, we explore the application of LIME to our proposed model to provide insights into the generated results.

## Experimental setup

The methods were developed and prototyped using the cloud-based Jupyter Notebook environment (Colab Notebook). The decision to utilize this environment was influenced by the abundance of freely available and suitable libraries for machine learning models, such as Scikit-learn, Matplotlib, Keras, among others.

## Result analysis without applying NSGA-II

To demonstrate the significance of our proposed model, we initially evaluate the performance results of our six employed classifiers (*e.g.*, DT, SVM, KNN, ET, RF, and SEHM) without applying multi-objective feature selection, NSGA-II. Table 4 showcases the accuracy, precision, recall, F1-score, Jaccard score (J-score), Cohen's Kappa score (CK-score), and Hamming loss (H-loss) of these classifiers. Accuracy assesses the model's correctness by quantifying the percentage of accurate classifications it achieves. Recall gauges the model's ability to accurately identify positive instances, whereas precision assesses the model's capacity to generate accurate positive predictions. The F1-score is a balanced measure of the model's overall performance that includes recall and precision. The J-score gauges the similarity between the predicted and actual classification sets, while the CK-score assesses the consistency between expected and actual classifications,

**Table 3 List the adjusted parameters for the classifiers that were used using GridsearchCV.**

| Classifiers | Optimal perameters |
|---|---|
| DT | criterion = gini, max depth = 10, splitter = best, random state = 15 |
| SVM | C = 1.0, kernel = linear, degree = 3, random state = 25 |
| KNN | n neighbors = 5, weights = uniform, leaf size = 25, algorithm = brute |
| RF | n estimators = 5, random state = 20, max depth = 3, max features = sqrt |
| ET | n estimators = 20, criterion = gini, max depth = 7, max features = sqrt |
| SEHM | estimators = RF, cv = iterable, stack method = predict proba |

**Table 4 Evaluate the different performance metrics for our employed classifiers without NSGA-II.**

| Metrics | DT | SVM | KNN | RF | ET | SEHM |
|---|---|---|---|---|---|---|
| Accuracy | 0.8376 | 0.8205 | 0.8547 | 0.9145 | 0.8717 | 0.9316 |
| Precision | 0.8095 | 0.8032 | 0.8867 | 0.9285 | 0.8307 | 0.9464 |
| Recall | 0.8793 | 0.8448 | 0.8103 | 0.8965 | 0.9310 | 0.9137 |
| F1-score | 0.8429 | 0.8235 | 0.8468 | 0.9122 | 0.8780 | 0.9298 |
| J-score | 0.7285 | 0.70 | 0.7343 | 0.9122 | 0.7826 | 0.8688 |
| CK-score | 0.6754 | 0.6411 | 0.7091 | 0.8289 | 0.7438 | 0.8631 |
| H-loss | 0.1623 | 0.1794 | 0.1452 | 0.0854 | 0.1282 | 0.0683 |

considering the chance agreement. Finally, the H-loss quantifies the average fraction of misclassified labels assigned to each sample.

This table demonstrates that our proposed SEHM achieves the highest results compared to the conventional classifiers for all kinds of performance metrics. In terms of accuracy, it performed 93.16% robust results, with only 6.83% H-loss. Additionally, obtained a 94.64% outstanding precision score, which indicates it has an excellent capability to produce accurate positive predictions. When we consider the results of the conventional model, RF outperformed the other models and performed 91.45% accuracy. That's why we utilized RF while developing our proposed SEHM, as mentioned in the previous section. In contrast, SVM obtained an overall lowest result of 82.05% accuracy with the highest 17.94% losses.

Subsequently, to illustrate the trade-off between the true positive rate and false positive rate across different threshold values, we evaluate the ROC curve for all employed models without NSGA-II. In Fig. 2, we display the completed ROC curves, with the true positive and false positive rates shown on the y and x-axes, respectively. Based on the results obtained, our proposed SEHM demonstrated outstanding performance, achieving ROC scores close to 98%. In contrast, the DT classifier exhibited lower scores, reaching around 87% score.

### Result analysis with applying NSGA-II

It is crucial to incorporate a multi-objective algorithm such as NSGA-II, which facilitates the effective investigation of intricate, conflicting goals by offering well-distributed Pareto-

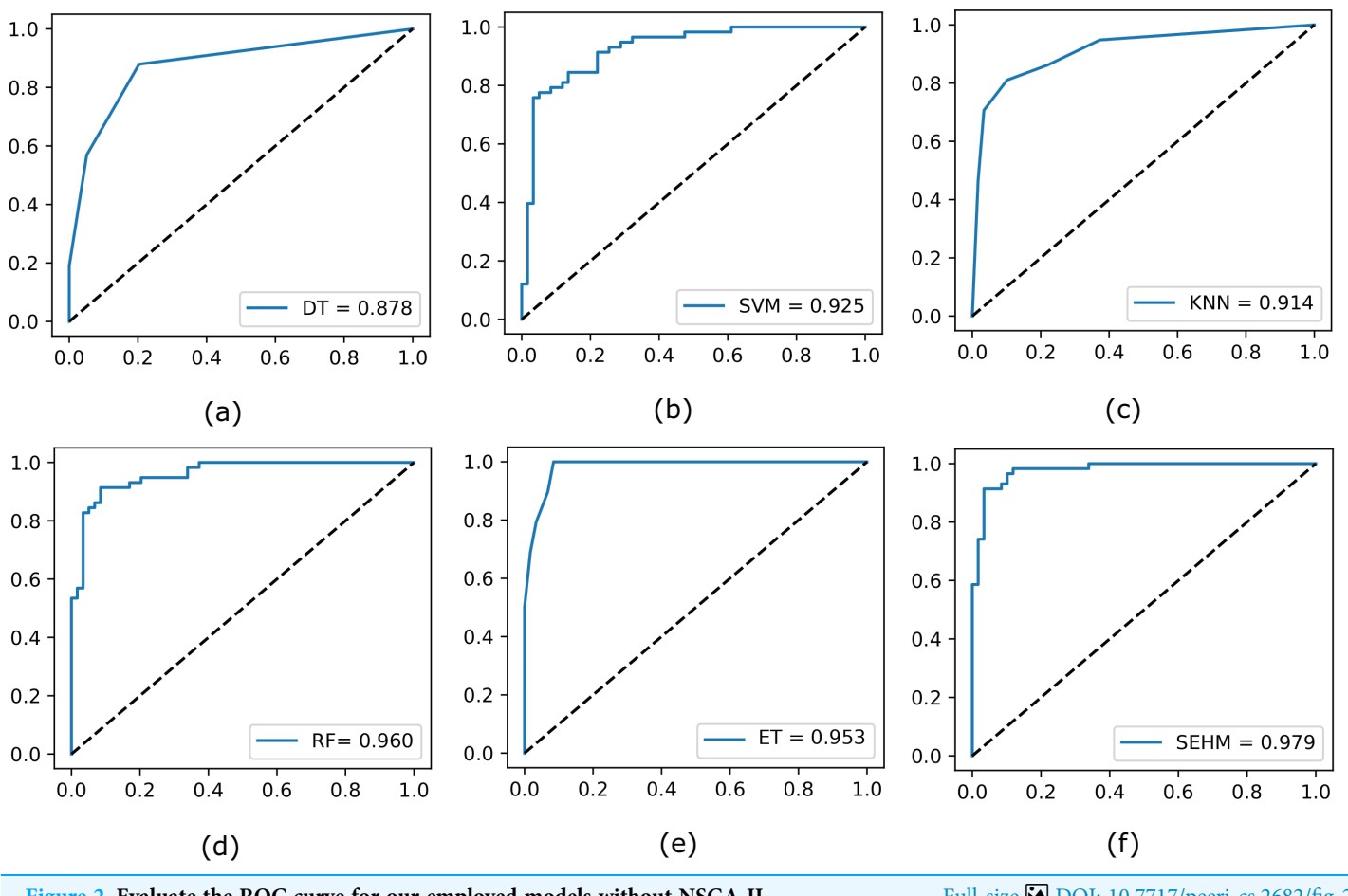

**Figure 2** Evaluate the ROC curve for our employed models without NSGA-II.

optimal solutions. By properly balancing several objectives and tackling real-world issues that single-objective approaches are unable to (without NSGA-II), this strategy provides decision-makers with a range of feasible solutions. To demonstrate the significance of a multi-objective feature selection, we customize NSGA-II with our proposed and employed conventional models. Initially, we represent the outcome of the PF, which helps to identify the set of non-dominated solutions in multi-objective optimization problems. Figure 3 illustrates the PF obtained from the employed models after applying NSGA-II, where MO-DT, MO-SVM, MO-KNN, MO-RF, MO-ET, and MO-SEHM refers to the multi-objective optimization for DT, SVM, KNN, RF, ET, and SEHM. Examination of the figures reveals that our proposed MO-SEHM obtained the highest accuracy of 94.87% with nine features. Considering only seven, six, five and four features, this model also performed a robust accuracy of 94.01%, 93.16%, 92.30%, and 91.45%, respectively. On the other hand, there is a remarkable improvement observed in the case of the performance of traditional classifiers, especially ET, demonstrating the significance of applying a multi-objective feature selection.

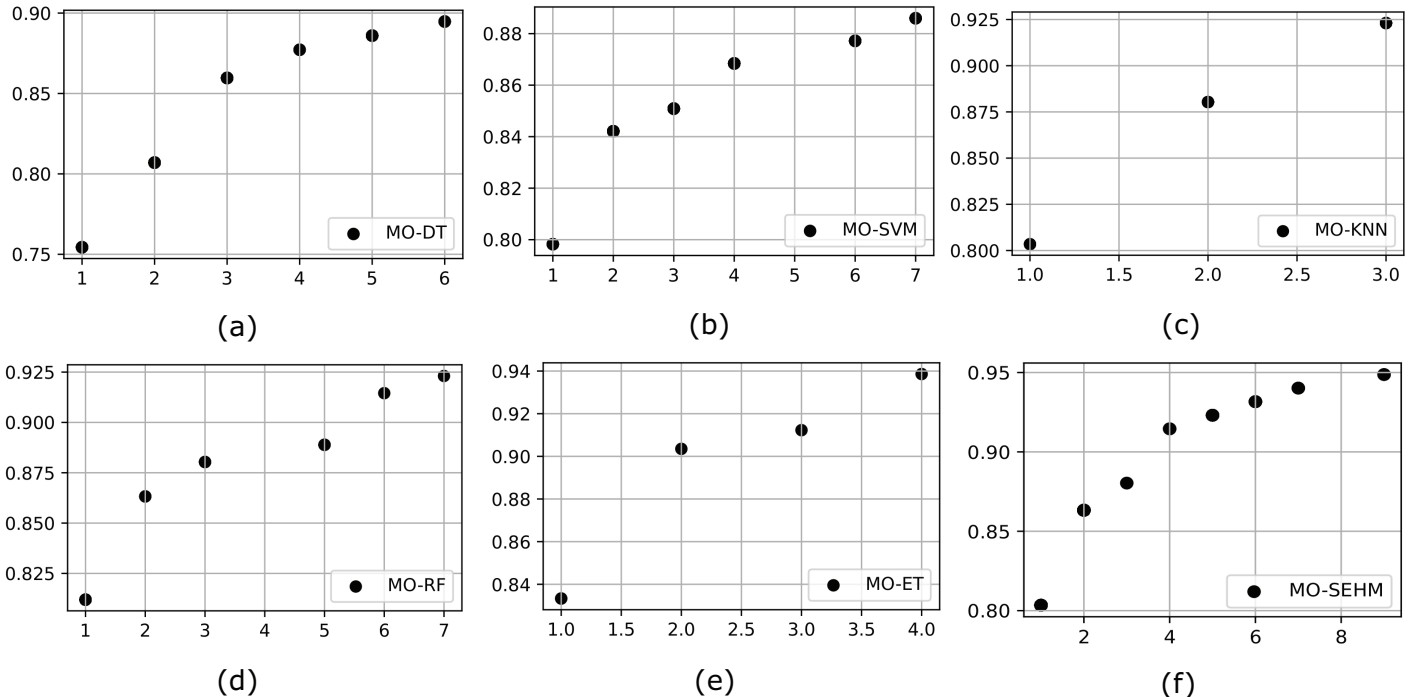

**Figure 3** Pareto front for our proposed and traditional models, where x and y-axis represent the minimum number of feature sets and maximum number of accuracy, respectively.

**Table 5** Evaluate the different performance metrics for our employed classifiers by applying NSGA-II.

| Metrics | MO-DT | MO-SVM | MO-KNN | MO-RF | MO-ET | MO-SEHM |
|---------|-------|--------|--------|-------|-------|---------|
| Accuracy | 0.8947 | 0.8859 | 0.9230 | 0.9230 | 0.9385 | 0.9487 |
| Precision | 0.8688 | 0.8666 | 0.9152 | 0.9313 | 0.9491 | 0.9482 |
| Recall | 0.9298 | 0.9122 | 0.9310 | 0.9103 | 0.9211 | 0.9482 |
| F1-score | 0.8983 | 0.8888 | 0.9230 | 0.9206 | 0.9348 | 0.9482 |
| J-score | 0.8153 | 0.80 | 0.8571 | 0.8604 | 0.8972 | 0.9016 |
| CK-score | 0.7894 | 0.7719 | 0.8461 | 0.8380 | 0.8501 | 0.8974 |
| H-loss | 0.1052 | 0.1140 | 0.0769 | 0.0769 | 0.0614 | 0.0512 |

Additionally, we evaluated the previous performance indicator metrics after applying NSGA-II, Table 5 showcases these outcomes. Our proposed MO-SEHM consistently outperforms applying NSGA-II compared to conventional classifiers across all performance metrics. With a robust accuracy of 94.87% and a minimal H-loss of 5.12%, MO-SEHM demonstrates the superiority of both NSGA-II and SEHM. Moreover, it achieves an outstanding precision, recall, and f1-score of 94.82%, indicating its ability to make accurate positive and balanced predictive outcomes. In comparison, among the conventional models, MO-ET stands out with an accuracy of 93.85%. Conversely, MO-
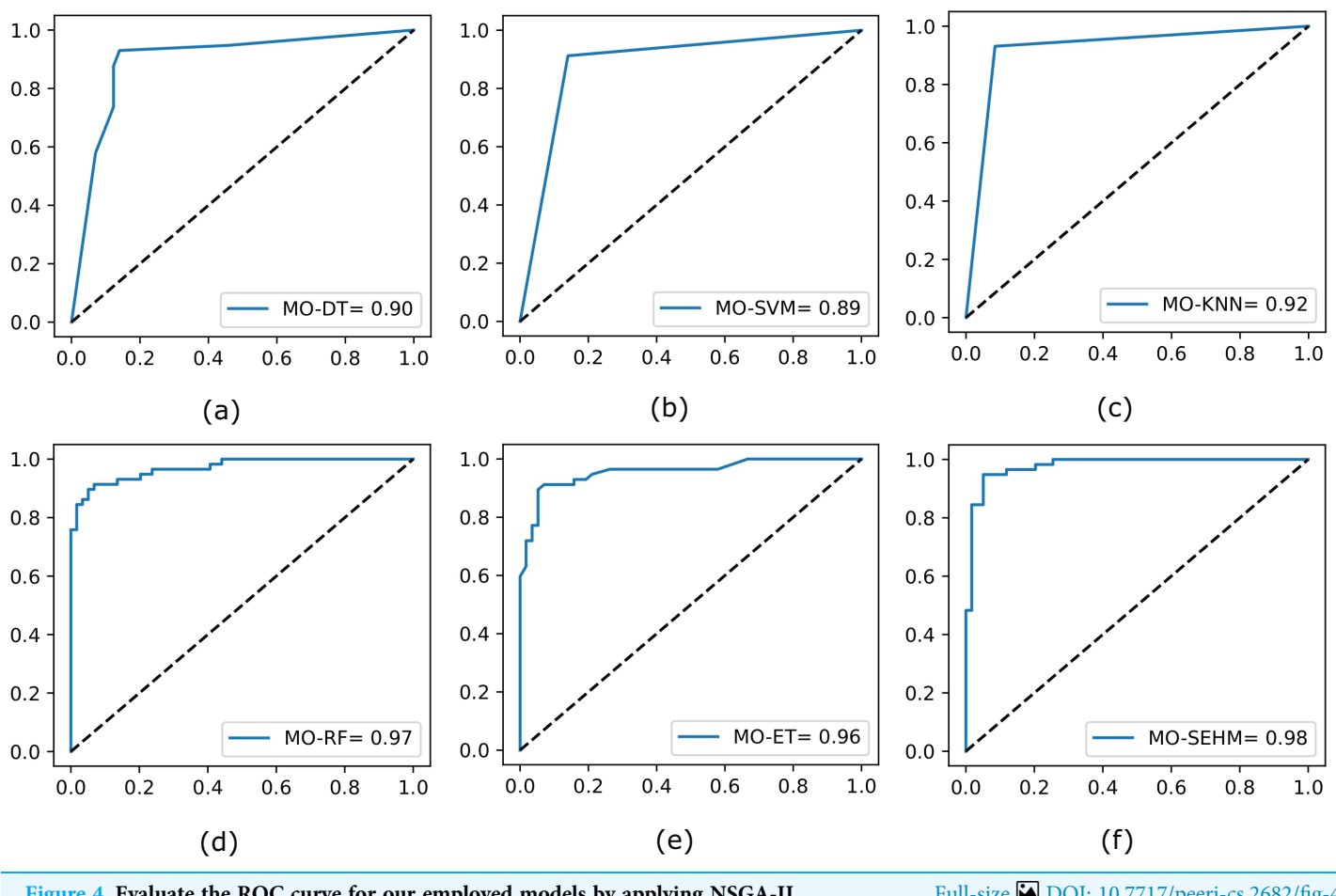

**Figure 4 Evaluate the ROC curve for our employed models by applying NSGA-II.**

SVM exhibits the lowest overall performance, achieving only 88.59% accuracy with an H-loss rate of 11.40%. Overall, we observed significant improvements in the performed results of all employed models by integrating multi-objective feature selection, NSGA-II.

Finally, we again assess the ROC curves for all utilized models by applying NSGA-II, shown in Fig. 4, where the true positive rate is plotted on the y-axis and the false positive rate is plotted on the x-axis. Our proposed MO-SEHM shows remarkable performance, with ROC scores of 98%. Conversely, the MO-SVM displays lower scores, reaching approximately 89%. On the other hand, RF and ET classifiers demonstrate effective scores, achieving around 97% and 96% respectively. These scores are comparatively superior to the previous ROC scores obtained without applying NSGA-II.

## Enabling explanable AI

Explainable AI refers to the set of methods used to make ML systems more transparent and interpretable, which is crucial, especially in domains where the decisions impact human lives directly, such as healthcare. Explainable AI provides insights into how the model arrives at a particular prediction or decision, allowing clinicians or end-users to trust the

**Table 6 A list of optimal feature sets considering the objective (maximize or optimal accuracy (OP-ACC)) by applying NSGA-II with our employed models.**

| Models | Anaemia | Time | Cr-Ph | Ser-C | Ej-Fr | Age | Diabetes | H-BP | Platelets | Ser-S | Sex | Smoking | OP-ACC |
|---|---|---|---|---|---|---|---|---|---|---|---|---|---|
| MO-DT | (✗) | (✓) | (✓) | (✓) | (✗) | (✗) | (✓) | (✗) | (✗) | (✗) | (✓) | (✓) | 89.47% |
| MO-SVM | (✓) | (✓) | (✓) | (✓) | (✗) | (✗) | (✗) | (✗) | (✗) | (✓) | (✓) | (✓) | 88.59% |
| MO-KNN | (✗) | (✓) | (✗) | (✗) | (✓) | (✗) | (✗) | (✗) | (✗) | (✓) | (✗) | (✗) | 92.30% |
| MO-RF | (✗) | (✓) | (✓) | (✓) | (✓) | (✓) | (✓) | (✗) | (✗) | (✗) | (✓) | (✗) | 92.30% |
| MO-ET | (✗) | (✓) | (✗) | (✓) | (✓) | (✗) | (✗) | (✗) | (✗) | (✓) | (✗) | (✗) | 93.85% |
| MO-SEHM | (✗) | (✓) | (✓) | (✓) | (✓) | (✓) | (✓) | (✗) | (✓) | (✓) | (✗) | (✓) | 94.87% |

model's recommendations and understand its limitations. Hence, we utilize an explainable AI method, named Local Interpretable Model-agnostic Explanations (LIME), and leverage our proposed model. It works by generating locally interpretable explanations for individual predictions made by complex models. Initially, we attempted to take the most relevant features considering our primary objective (maximize accuracy) after applying NSGA-II. Table 6 holds the list of these optimal features selected by our employed models. This table illustrates that MO-SEHM achieves the highest accuracy of 94.87% with nine features. These features are time, creatinine phosphokinase (Cr-Ph), serum creatinine (Ser-C), ejection fraction (Ej-Fr), age, diabetes, platelets, serum sodium (Ser-S), and smoking. Hence we applied LIME on these features and trained using the SEHM model to generate the predictions.

Table 7 presents the prediction probabilities for two randomly selected data samples, one positive and one negative. The "Actual value" column indicates the actual values of each feature, while the "Negative reasons" and "Positive reasons" fields show the LIME-generated values, indicating whether a feature has a negative or positive influence on prediction probabilities. For example, if a feature negatively affects a sample, its name and recommended value ranges are filled in the "Negative reasons", while a positive influence is indicated in the "Positive reasons". In the case of a random positive sample, MO-SEHM predicts a 99% probability of having HF death. The feature "time" contributes most significantly to this positive prediction, with its actual value falling equal to or lower than the recommended range of 58. Other features such as "Ser-C," "Ser-S," "Cr-PH," and so on also contribute significantly to the positive prediction. Similarly, in the case of the Negative prediction, MO-SEHM predicts a 98% probability of not having HF death. Again "time" emerges as the most influential feature in predicting, its value of 212 falls within the recommended range of greater than 186. Additionally, other feature values such as "Cr-Ph," "Ej-Fr," "diabetes," "platelets," "age," "smoking," and "Ser-S" contribute to the negative prediction.

Moreover, we also considered applying another interpretable AI named Partial Dependence Plot (PDP) to explore the global behavior of the most optimal feature set selected by MO-SEHM. PDP is a straightforward and model-agnostic approach to understanding how the independent features behave according to its target variable. It visualizes the correlation between an independent and dependent feature by randomly

**Table 7 Outcome explanations generation by LIME for a random positive and negative case of the HF using MO-SEHM.**

| Probability of positive prediction (99%) | | | Probability of negative prediction (98%) | | |
|---|---|---|---|---|---|
| Negative reasons | Positive reasons | Actual value | Negative reasons | Positive reasons | Actual value |
| (−) | time <= 58 | 30 | Time > 186 | (−) | 212 |
| (−) | Ser-C > 1.69 | 3.50 | Cr-Ph < 90 | (−) | 81 |
| (−) | Ser-S <= 134 | 132 | – | 1.18 > Ser-C | 1.3 |
| 204,000 < platelets < 304,500 | (−) | 228,000 | Ej-Fr < 30 | (−) | 21 |
| (−) | 298 < Cr-Ph < 651 | 582 | Diabetes <= 0 | (−) | 0 |
| (−) | 30 <= Ej-Fr | 35 | Platelets < 304,500 | (−) | 233,000 |
| (−) | Smoking > 0 | 1 | 37.71 < age <= 50.00 | (−) | 43 |
| (−) | 0 < diabetes <= 1 | 0 | Smoking <= 0 | (−) | 0 |
| (−) | 60 < Sex <= 71.39 | 69 | 136 < Ser-S <= 149 | (−) | 139 |

shuffling each feature's values and observing the resulting impact. Figure 5 depicts the PDP of time, Cr-Ph, Ser-C, Ej-Fr, age, diabetes, platelets, Ser-S, and smoking features. Where the y-axis represents the partial dependence score (PDS) and the x-axis indicates the actual value of a feature. PDS is the average expected value of the target variable when all other features are held constant and the feature of interest is varied across its range. By illustrating how feature modifications affect predictions, this score offers valuable information on the significance of features and the types of feature interactions that occur inside the model. The minor ticks in the x-axis indicate the different values of a feature and the black line in each plot is the PDP line. When this line is comparatively high for the particular value range, it suggests that this value range is susceptible to HF mortality. The time plot demonstrates that within a small follow-up period, most HF patients died, and when increasing the period, this rate decreases. From the Cr-Ph plot, it is apparent that a lower value of Cr-Ph is risky to HF patients. On the other hand, Ser-C illustrates that a higher value is abnormal for the patients. The Ej-Fr plot indicates a lower value of less than 30 is risky for HF patients. Then the age plot demonstrates that older patients are more vulnerable to HF death. Subsequently, platelets and Ser-S plots indicate having a lower value of these characteristics contains a high mortality risk. Finally, the categorical variables, diabetes and smoking, showcase one or more positive values susceptible to HF mortality.

# DISCUSSION

## Interpretation of the results

The growing demand for superior healthcare services underscores the importance of ML techniques in the medical sector. Our proposed model, MO-SEHM, which combines multi-objective feature selection using NSGA-II with the Stacked Enable Hybrid Model (SEHM), demonstrates significant improvements in healthcare predictions. Specifically, the experimental results highlight its capability in early-stage HF mortality prediction by

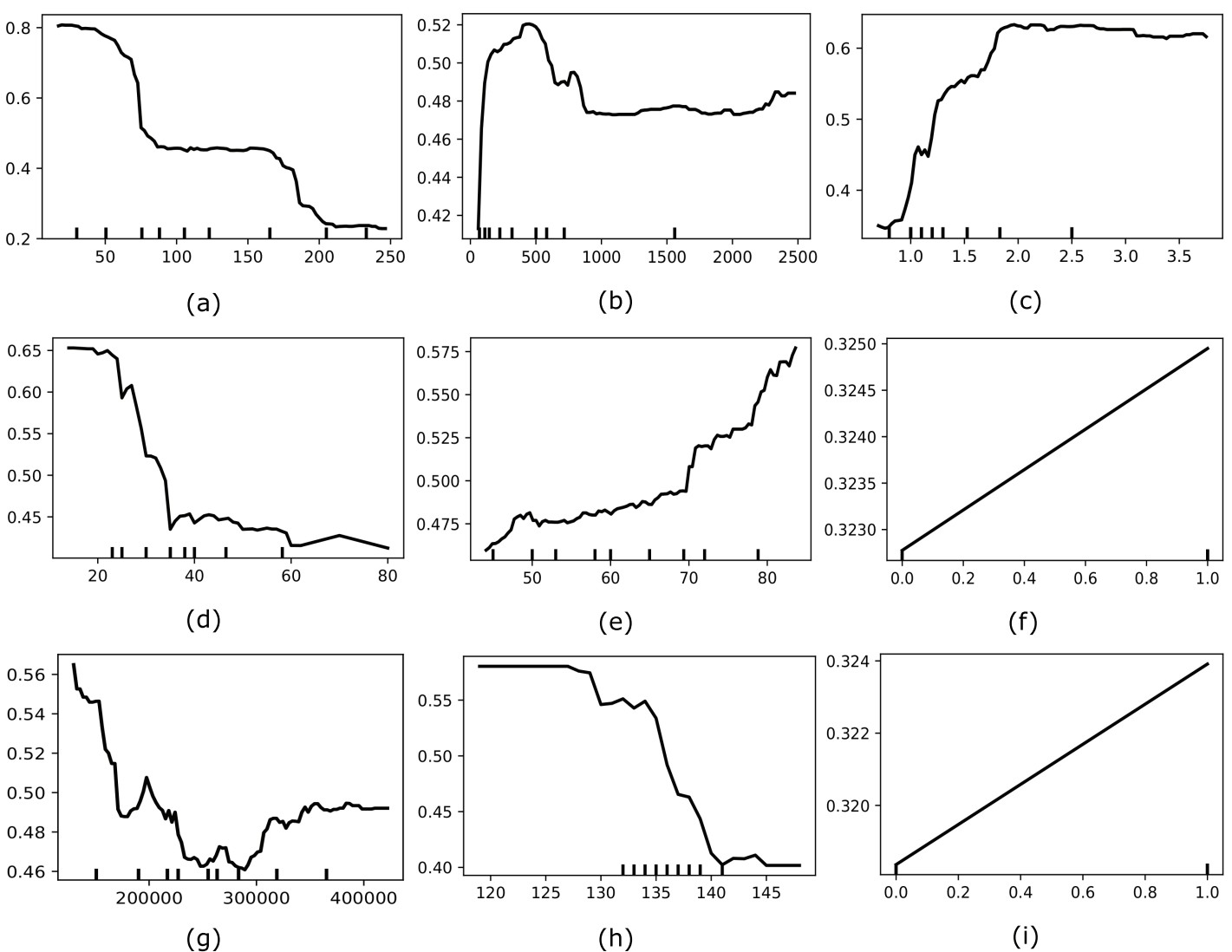

**Figure 5** Showcases the PDP of most optimal features for our model (*e.g.*, (A) time, (B) Cr-Ph, (C) Ser-C, (D) Ej-Fr, (E) age, (F), diabetes, (G), platelets, (H) Ser-S, and (I) smoking).

achieving superior scores across various classification metrics, including an accuracy rate of 94.87%.

Moreover, the integration of ensemble learning and multi-objective optimization techniques in MO-SEHM effectively addresses common challenges, such as overfitting and underfitting (*Soui et al., 2021b*). The SMOTE and TomekLink methods employed for addressing class imbalance further contributed to the model's robust performance (*Hairani, Anggrawan & Priyanto, 2023*). By ensuring a balanced distribution of data, the model showcased improved generalizability and reduced overfitting risks (*Sutradhar et al., 2023a*). The comparative analysis, as depicted in Table 8, highlights the superiority of MO-SEHM over existing state-of-the-art approaches, offering valuable insights for future

**Table 8 Performance comparison between our approach and existing ML-based HF mortality diagnosis methods.**

| Study | Data source | Data balancing method | Multi-objective optimization | Performed model | Achieve accuracy | Explainable AI |
|---|---|---|---|---|---|---|
| *Newaz, Ahmed & Haq (2021)* | FIOC | SMOTE | (✗) | RF | 76.25% | (✗) |
| *Ishaq et al. (2021)* | FIOC | SMOTE | (✗) | ET | 92.62% | (✗) |
| *Le et al. (2021)* | UCI | (✗) | (✗) | RF | 74% | (✗) |
| *Hussain et al. (2021)* | Physionet databases | (✗) | (✗) | SVM | 88.79% | (✗) |
| *Sutradhar et al. (2023b)* | FIOC | SMOTE-ENN | (✗) | IBS | 92.75% | (✗) |
| *Li et al. (2022)* | The eICU-CRD (v-0.2) | (✗) | (✗) | XGB | 82.6% | SHAP |
| *Mishra (2022)* | FIOC | SMOTE | (✗) | SVM | 83.33% | (✗) |
| *Plati et al. (2021)* | IUHI | SMOTE | (✗) | ROT | 91.23% | (✗) |
| *Nishat et al. (2022)* | FIOC | SMOTE-ENN | (✗) | RF | 90% | (✗) |
| *Sabahi, Vali & Shafie (2023)* | PRCVD | SMOTE | (✗) | XGB | 76.4% | (✗) |
| *Luo et al. (2022)* | MIMIC | (✗) | (✗) | XGB | 83.1% | (✗) |
| *Chicco & Jurman (2020)* | FIOC | (✗) | (✗) | RF | 74% | (✗) |
| *Mpanya et al. (2023)* | PMRCardio Database | (✗) | (✗) | RF | 88% | (✗) |
| *Mohan, Thirumalai & Srivastava (2019)* | UCI | (✗) | (✗) | HRFLM | 88.7% | (✗) |
| *Rahman et al. (2023)* | Physionet | (✗) | (✗) | Stacking | 89.41% | (✗) |
| *Sutradhar et al. (2023c)* | FIOC | BOO-ST | (✗) | CBCEC | 93.67% | (✗) |
| **This study** | **FIOC** | **SMOTE and Tomeklink** | **NSGA-II** | **SEHM** | **94.87%** | **LIME** |

advancements in the domain. Where the sign (✗) indicates that this study does not consider the specific method or technique, otherwise fill the row with the name of the utilized method. The University of California Irvine, Ireland, and University Hospital of Ioannina, Persian Registry of Cardio Vascular disease, and Medical Information Mart for Intensive Care refers to a short form in the table of UCI, IUHI, PRCVD, and MIMIC, respectively.

## Implications

The findings of this study have significant implications for clinical decision-making and predictive modeling in the healthcare sector. MO-SEHM has the potential to alleviate the burden on healthcare professionals by automating and enhancing processes such as early diagnosis and mortality prediction. Its capability to achieve high accuracy and generalization suggests its applicability in real-world medical settings, enabling timely interventions and better patient outcomes.

Furthermore, the use of multi-objective feature selection offers a systematic approach to identifying relevant features, which not only improves predictive performance, but also reduces computational complexity. This approach can be extended to other healthcare datasets to address similar challenges, making the proposed methodology a versatile tool

for various clinical applications. By leveraging advanced techniques like NSGA-II, this study introduces a scalable framework that aligns with the growing need for precision in modern healthcare.

## Limitations

Despite its advantages, our proposed MO-SEHM model faces certain limitations. The integration of NSGA-II with SEHM results in a complex architecture, which increases computational costs and implementation difficulties. This complexity may limit the adoption of the model in resource-constrained environments. Additionally, the evaluation of the model was conducted on a relatively small dataset from the FIOC, which can lead to poor generalization, high variance in results, and increased overfitting risks. While the methods employed, such as SMOTE and ensemble learning, mitigate some of these challenges, the dataset size remains a limiting factor in providing comprehensive insights into the model's performance across diverse patient populations. Addressing these limitations is crucial to ensuring the reliability and effectiveness of the hybrid approach in broader applications.

## Future work

In future research endeavors, we aim to address the limitations identified in this study. Firstly, we plan to evaluate MO-SEHM on larger, more diverse datasets to enhance its generalizability and robustness. By leveraging data from multiple sources and regions, we can mitigate biases and improve the model's applicability across various healthcare settings. Moreover, our objective is to explore other hybrid ML models that offer better precision and scalability. These models will be assessed for their effectiveness in handling diverse datasets and domains, with a focus on reducing computational costs while maintaining high performance.

Additionally, we envision integrating our proposed approach into a blockchain network to enhance the security and accessibility of healthcare information. This integration can facilitate seamless information exchange across hospitals and clinics, ensuring data privacy and promoting collaborative decision-making. Future work will also involve the exploration of federated learning frameworks to enable decentralized data sharing while maintaining patient confidentiality, further expanding the utility of our methodology in real-world scenarios.

## CONCLUSION

Despite significant progress in the medical field, clinicians encounter increasing challenges in reducing the incidence of mortality due to HF. To address this increasing influence, we have introduced an effective ML-based early warning system for heart failure mortality. The experimental analysis underscores the pivotal role of the proposed method, particularly when integrating the multi-objective feature selection, NSGA-II. Additionally, predictive outcomes of every employed model are significantly enhanced by applying it, specifically, our proposed SEHM achieves a remarkable accuracy over conventional models. Finally, applying LIME has improved the reliability of our model, interpreted

explanations for model predictions, and assisted physicians in comprehending the components influencing each outcome, thereby enabling practical applicability. This clarity fosters trust and makes it possible to make better-informed patient-specific treatment decisions.

### Funding
The authors received no funding for this work.

### Competing Interests
The authors declare that they have no competing interests.

### Author Contributions
- F M Javed Mehedi Shamrat conceived and designed the experiments, performed the experiments, analyzed the data, performed the computation work, prepared figures and/or tables, authored or reviewed drafts of the article, and approved the final draft.
- Majdi Khalid conceived and designed the experiments, performed the experiments, analyzed the data, performed the computation work, authored or reviewed drafts of the article, and approved the final draft.
- Thamir M. Qadah conceived and designed the experiments, performed the experiments, analyzed the data, authored or reviewed drafts of the article, and approved the final draft.
- Majed Farrash analyzed the data, authored or reviewed drafts of the article, and approved the final draft.
- Hanan Alshanbari analyzed the data, authored or reviewed drafts of the article, and approved the final draft.

### Data Availability
The data is available at Kaggle: https://www.kaggle.com/datasets/andrewmvd/heart-failure-clinical-data.

The code is available at Zenodo: F M Javed Mehedi Shamrat. (2024). Shamrat777/Multi-Objective-Hybrid-Model-for-HF-Mortality: v1.0 (v1.0). Zenodo. https://doi.org/10.5281/zenodo.14211501.

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
