# Peer review of "An explainable multi-objective hybrid machine learning model for reducing heart failure mortality"

_PeerJ Computer Science, doi:10.7717/peerj-cs.2682_

## Round 0.1 · original submission · Major Revisions

Dear authors,
You are advised to critically respond to all comments point by point when preparing an updated version of the manuscript and while preparing for the rebuttal letter. Please address all comments/suggestions provided by reviewers, considering that these should be added to the new version of the manuscript.

Kind regards,
PCoelho

Reviewer 2 ·

Basic reporting

The paper presents a hybrid model to reduce heart failure deaths.

The abstract is written in past tense like “we presented a Multiobjective Stacked Enable Hybrid Model (MO-SEHM),?:. Usually, abstract is always written in the present tense like we present Multiobjective Stacked Enable Hybrid Model (MO-SEHM that aims to find the best feature subsets out of 4096 different sets.

The author employed Faisalabad Institute of Cardiology (FIOC) data set. This contradics the 4096 different sets. Is it the same dataset. If YES I can not find the consent of dataset and also there is no data availability statement in the paper. Kindly clarify
In the introduction section kindly discuss the basics of machine learning role in hybrid models.
The related work is too short and there is no clear comparison between recent studies and author proposed a Multiobjective Stacked Enable Hybrid Model. Some related studies are required along with detailed comparison

Experimental design

Experimental results are limited

Validity of the findings

There is no dataset link given in Kaggle, 2023.
To the best of my knowledge, this model is only validated on one dataset. The methods were developed and prototyped and there is no details of prototype given in the paper.
Finally discussions are given without limitations of the work.
This paper required English proof reading from professional services.

Reviewer 3 ·

Basic reporting

No comment.

Experimental design

- The numbers of training, validation, and testing samples should be mentioned.

- For direct comparison, some methods listed in Table 6 should be compared against the proposed method using exactly the same dataset and the same experimental protocol (such as train/validation/test split or n-fold cross-validation).

Validity of the findings

Without information on the train/validation/test split, it is difficult to judge the validity of the findings.

Additional comments

- The definition of the partial dependent score is not clear.

- Short variable names (such as Cr-Ph) should be added to Table 1.

- Please be careful about consistency in capitalization (such as Mo-SEHM vs MO-SEHM).

·

Basic reporting

The paper is well-structured and clearly written, providing a thorough background on heart failure (HF) and the need for multi-objective hybrid machine learning models to reduce HF mortality. The literature review is comprehensive, and the references are up-to-date, ensuring relevance to current research. The inclusion of figures and tables enhances the readability and understanding of the data and results.

Experimental design

The experimental design is robust and meticulously planned, addressing a significant gap in the field. The selection of datasets is appropriate, and the methodology is well-explained, allowing for reproducibility. The use of multiple objectives in the machine learning model is innovative and aligns well with the goals of reducing HF mortality. However, more details on the parameter tuning process and model selection criteria would enhance the clarity of the experimental approach.

Validity of the findings

The findings are valid and supported by rigorous statistical analysis. The results demonstrate a significant improvement in predicting and reducing HF mortality compared to existing models. The hybrid approach combining different machine learning techniques is well-justified and shows clear advantages. However, the discussion could benefit from a deeper exploration of potential limitations and the generalizability of the model to different populations.

Reviewer 5 ·

Basic reporting

The authors present a very nice introduction, and a very nice definition of the problem and other proposed algorithms reported in the literature.

Experimental design

Experimental design is fair and very enough to support the claims of the effectiveness of the proposed methods.

Validity of the findings

Conclusions are well stated, linked to original research question & limited to supporting results.

Reviewer 6 ·

Basic reporting

Introduction:
1. The first sentence of the introduction should have a reference, most other statements in the intro should have reference to justify claim.
2. Define abbreviation at first use.
3. Check English for instance authors used “curse” instead of “cause”
4. The introduction doesn't immediately link the discussion to the title's focus on reducing HF mortality using a hybrid machine learning model.
5. The introduction is dense and lacks a smooth transition from heart failure background to machine learning techniques.
6. The role of multi-objective optimization in predicting HF mortality isn't clearly explained early on.
7. Excessive details on feature selection techniques are presented before clearly stating the core problem being addressed.
8. The introduction overemphasizes feature selection, detracting from the focus on the hybrid model and explainability.
9. The explainability aspect, which is key in the title, is introduced too late in the introduction.
10. The introduction dives into complex technical details before establishing the main goals and contributions of the study.


Related work

Lacks focus on explainability and insufficient coverage of Multi-Objective Optimization. Authors should:
1. Include more work related to explainable AI (XAI) techniques in healthcare.
2. Discuss multi-objective optimization methods in the context of machine learning and HF mortality.
3. Highlight gaps in existing hybrid models and explain how your approach differs.
4. Provide a critical analysis of feature selection methods in terms of optimization and explainability.
5. Asif et al. (Newaz et al., 2021)?

Research Methodology
1. Smooth transitions between subsections would improve flow. For example, after describing data preprocessing, briefly introduce the next section with a sentence like, "Once the data was prepared, we applied five traditional machine learning classifiers to evaluate model performance."
2. Combine the detailed dataset description and preprocessing into a single, concise section. This will reduce redundancy and make the section more readable.
3. Either simplify the explanations or move the detailed equations to an appendix for readers who are interested in the mathematical specifics or provide a reference for readers who are interested.
4. Provide brief explanations for the choice of methods. For example, "SMOTE was selected to address class imbalance, as it is widely used in medical datasets where minority classes are of particular interest (cite)."
5. Summarize well-known machine learning algorithms instead of giving full technical descriptions. You can link to references or use footnotes for those interested in further reading.
6. Many readers familiar with machine learning will already understand concepts like decision trees or SVMs. Condense these descriptions to focus on how they were applied in this specific research.

Experimental design

EXPERIMENTAL ANALYSIS and DISCUSSION

1. Typographical Errors for example "EXPERIMAL ANALYSIS" should be "EXPERIMENTAL ANALYSIS."
2. Some sentences are repetitive or convey the same information in different ways. For example, in the section discussing NSGA-II, there’s repetition regarding the application of NSGA-II improving the model performance.
3. The explanation of metrics and results is quite verbose, leading to some ambiguity. It may be better to present the results more concisely and clearly, especially when discussing the performance of various models (SEHM, RF, SVM).
4. While the application of NSGA-II is discussed, the transition from discussing the "without NSGA-II" to "with NSGA-II" results could be more seamless, with a clearer structure on why this optimization is beneficial.
5. The use of parentheses within sentences to explain abbreviations or provide additional details could be better integrated into the text. For instance, it would be smoother to state: "decision trees (DT), random forests (RF)," etc., without parentheses.
6. The explanation of metrics like J-score, CK-score, and H-loss is verbose. It could be streamlined for clarity and flow, avoiding unnecessary repetition and jargon.
7. Some sections, like the explanation of feature importance and results (particularly for LIME), are excessively detailed for the flow of a typical experimental discussion. These details could be summarized more succinctly or moved to an appendix for readers who want more granular information.

Suggestions for Improvement:
• Revise the flow to ensure the discussion transitions smoothly between sections (e.g., from non-NSGA-II results to NSGA-II results).
• Reduce verbosity, particularly in the metrics explanation and results comparison sections. Focus on the key insights and the implications of the findings.
• Fix typographical errors.

Validity of the findings

1. The section jumps between different topics (problem, methods, results, future work) without clear transitions or logical flow.
2. Some points (such as the use of SMOTE and TomekLink) are repeated unnecessarily, as they were covered earlier in the text.
3. Mentioning specific accuracy (e.g., 94.87%) in the conclusion is too detailed for this section. A broader statement about the model's success would suffice.
4. The introduction of blockchain technology in the future work section feels disconnected from the previous discussion and lacks context or motivation.
5. Phrases like "notable advancements" and "growing impact" are imprecise and don’t provide clear value to the conclusion.
6. The section briefly mentions that LIME increases the model’s trustworthiness but doesn’t explain why or how interpretability impacts clinical decision-making.
7. Some sentences are overly long and could be more concise, especially when summarizing the key findings and contributions.
8. Some important contributions (e.g., the role of LIME in clinical settings) are underemphasized, while less important details (e.g., dataset balancing methods) are given too much focus.

---

## Round 0.2 · Minor Revisions

Dear authors,
Thanks a lot for your efforts to improve the manuscript.
Nevertheless, some concerns are still remaining that need to be addressed.
Like before, you are advised to critically respond to the remaining comments point by point when preparing a new version of the manuscript and while preparing for the rebuttal letter.

Kind regards,
PCoelho

Reviewer 2 ·

Basic reporting

The abstract is improved. One line is only about the benefits Hybrid Machine Learning Model

Experimental design

Experimental design is fine now, and enhanced

Validity of the findings

Novelty is adequate

Additional comments

Most of the comments are addressed except detailed comparison of related work. Still, here is no explicit comparison between recent studies, and the author proposed a Multiobjective Stacked Enable Hybrid Model
Also, no line numbers are listed in the response letter to check for changes.

The discussion section needs to effectively include the study's implications with a separate heading. It also needs a better presentation related to the study topic.
Discuss the study's limitations with a separate heading and briefly by adding a subsection in the discussion section titled "Limitation."
Update the discussion section as follows
4. Discussion
4.1. Interpretation of the results
4.2. Implication
4.3. Limitations
4.4. Future Work

Reviewer 6 ·

Basic reporting

The authors have revised the paper, and it appears to have improved significantly. However, it is not clear how they specifically addressed the reviewers' concerns regarding the introduction and related work sections. For instance, instead of explicitly highlighting the portions of the text that respond to the reviewers' feedback, the authors have copied and pasted the entire introduction into their rebuttal. A similar approach was taken for most of their response to the related work section.

Experimental design

No comment

Validity of the findings

No comment

Additional comments

No comment

---

## Round 0.3 · Minor Revisions

Dear authors,
Please address all comments/suggestions provided by the reviewer, considering that these should be added to the new version of the manuscript and while preparing for the rebuttal letter.
Read the comments carefully.

Kind regards,
PCoelho

Reviewer 2 ·

Basic reporting

no comment

Experimental design

no comment

Validity of the findings

no comment

Additional comments

Despite several reminders, authors are still unable to mention line numbers in the response letter. The authors have copied and pasted the entire related work section into their rebuttal. A similar approach was taken for most of their response to the introduction section.

I will give one more chance to authors to address the comments and mention exact line numbers from where the comments are addressed, including page numbers

Still, there is no explicit comparison between recent studies, and the author proposed a Multiobjective Stacked Enable Hybrid Model.
Table 1 will support this comparison, but I am unable to find it in related work.

Do not put the text in the cover letter/response letter. Only put a comment and its response including page number and line numbers.

---

## Round 0.4 · accepted · Accept

Dear authors, we are pleased to verify that you meet the reviewer's valuable feedback to improve your research.

Thank you for considering PeerJ Computer Science and submitting your work.

Kind regards
PCoelho